# Can Health Disparity Be Eliminated? The Role of Family Doctor Played in Shanghai, China

**DOI:** 10.3390/ijerph17155548

**Published:** 2020-07-31

**Authors:** Jiaoling Huang, Qin Zhu, Jing Guo

**Affiliations:** 1School of Public Health, Shanghai Jiao Tong University School of Medicine, Shanghai 200025, China; 2School of Social Development and Public Policy, Fudan University, Shanghai 200433, China; 17210730053@fudan.edu.cn

**Keywords:** health disparity, family doctor, system dynamics modeling, socioeconomic status, China

## Abstract

*Background*: Globally, the elimination of health disparity is a significant policy target. Primary health care has been implemented as a strategy to achieve this target in China for almost 10 years. This study examined whether family doctor (FD) policy in Shanghai contributed to eliminating health disparity as expected. Methods: System dynamics modeling was performed to construct and simulate a system of health disparity formation (business-as-usual (BAU) scenario, without any interventions), a system with FD intervention (FD scenario), and three other systems with supporting policies (Policy 1/Policy 2/Policy hybrid scenario) from 2013 to 2050. Health disparities were simulated in different scenarios, making it possible to compare the BAU results with those of FD intervention and with other policy interventions. Findings: System dynamics models showed that the FD policy would play a positive role in reducing health disparities in the initial stage, and medical price control—rather than health management—was the dominant mechanism. However, in this model, the health gap was projected to expand again around 2039. The model examined the introduction of two intervention policies, with findings showing that the policy focused on socioeconomic status improvement would be more effective in reducing health disparities, suggesting that socioeconomic status is the fundamental cause of these disparities. Conclusions: The results indicate that health disparities could be optimized, but not eliminated, as long as differences in socioeconomic status persists.

## 1. Introduction

Comparing the health status of civil servants at different levels, Marmot et al. found that men in the lowest grade had three times the mortality rate from coronary heart disease compared with the highest grade (administrators), from a range of other cause, and from all causes combined, and drew the conclusion that health status follows a certain social gradient [1]. The Black Report, published in 1980, proposed that health disparity was an important item for the academic and policy agenda for industrialized countries, defining health disparity in terms of the relation between the death rate and social class, the latter of which was measured by income, reputation, education, and other aspects [2]. Subsequent studies have confirmed the existence of varying degrees of health disparities on health outcome (e.g., self-reported health) among different social classes in both developed and developing countries [3,4]. Studies in the 2000s argued that the problem of health disparity has not been alleviated in developed countries by the development of the social economy and the advancement of medical technologies; rather, health disparity has actually worsened [5]. Over the next decade, a consensus has developed among scholars that economic development cannot alleviate health disparity but, on the contrary, widens the health gap among social classes defined by socioeconomic status (SES) [6]. SES was found to influence health through an individual’s negative emotions and cognitions, living environment, and lifestyle [7,8]. A range of resources have been included in conceptualizations of SES: These include money, knowledge, prestige, power, and beneficial social relationships [9]. Most scholars consider SES to be a key factor affecting health. Marmot argued that examining the gap between the upper and lower socioeconomic groups was a feasible method of describing the degree of inequality [10]. This phenomenon has not only been reported in developed countries. Survey data in developing countries, such as Bangladesh, Chile, and the Russian Federation, also showed that people with higher levels of education have lower mortality rates than do those with lower levels of education [11,12,13]. A study of the health status of the Chinese population found that groups with higher income also had better self-rated health status [14].

To deal with global health disparity, in the second half of the 1990s, countries across the world proposed policies to address health disparity. For example, in the United States, the Healthy People Initiative has repeatedly set the elimination of health disparities as a strategic policy target. In contrast to the traditional approach of pro-poor policies of disease management and specialty care, primary health care with its characteristics of lower cost, person- (rather than disease-) focused care, comprehensiveness of services available, and coordination (when care from other places is required) has been highlighted by authors from both the People’s Health Movement [15] and WHO [16], especially under the market orientation of a neo-liberal globalization background, which has worked against improvements in general and for equity in health specifically [17]. Global evidence from the USA, Canada, Thailand, and other countries showed that better primary care resources preferentially improve health more in socially disadvantaged populations than in the majority of the population [18,19,20,21]. Primary health care was listed as an ideal health care model in the 1978 Almaty Declaration and became the core concept of the World Health Organization’s (WHO) goal of achieving health for all [22]. However, in its initial stage, the Almaty Declaration was criticized as being overly idealistic. By 2008, in response to growing global health disparity, a WHO report called on all countries around the world to follow the Almaty Declaration, arguing that primary health care and the social determinants of health should have a significant place in public policy in all countries [23].

In China, from the 1960s to 1980s, “barefoot doctors”, who received basic health care training to improve population health by promoting low-cost services in rural areas, can be seen as primary health care in its embryonic form [24]. These individuals provided basic health services as doctors and participated in collective labor as farmers, and they related closely to their clients [25]. In rural environments where medical resources were scarce, barefoot doctors played a significant role in the prevention and treatment of frequently occurring diseases, making considerable contributions to health care in China [26]. However, with the reform of the agricultural and economic systems in the 1980s, the barefoot doctor system gradually collapsed. After economic reform in the 1980s, market incentive was introduced into healthcare institutions, especially tier-2 and tier-3 hospitals, resulting in profit-seeking behaviors of hospitals, a rapid growth of medical expenses, and the deficit of social medical insurance [27]. In response to the worsening problems of high costs of medical care and poor access to medical treatment in China, the Chinese government carried out a new round of reforms in 2009. The reform sought to enhance access to basic medical services for all residents by strengthening the construction of tier-1 hospitals, also called community health service centers (CHSCs) or stations. Family doctors (FDs) were primary care doctors who worked in CHSCs, as the government defined them. They are often called general practitioners or family physicians in other countries [28]. Since then, the family doctor system was established. In 2016, “Healthy China 2030” proposed to achieve universal health based on the FD system. The FD system is a system based on FDs who try to sign with residents in the administrative area, especially vulnerable groups of people, to provide FD-contracted services (including medical treatment, physical examination, health education, electronic health records creation and updates, family hospitalization, mental health consultation, etc.) and refer severe patients to specialists. FD-contracted services was initially a marketing strategy to attract residents to first-visit CHSCs; however, patients kept visiting tertiary hospitals without first-contact at community health service centers (CHSCs) [29]. This system also plays an active role in improving access to health services for vulnerable groups, such as those who are economically disadvantaged or have disabilities.

Existing studies have extensively portrayed the role of FDs as that of health gatekeepers [30]. Further, numerous studies have examined the role of FDs in improving unhealthy lifestyles, such as smoking and alcohol abuse, and in reducing or preventing the incidence of chronic diseases [31,32]. For example, Regan et al. conducted a comparative study of rural health service centers and the general rural population, finding that CHSCs not only improved rural residents’ access to medical services, but also reduced the disease risk of those served by these primary health care centers [33]. The role of FDs in determining health behaviors and health status has been widely discussed, but the influence of FDs in terms of contributing to the elimination of health disparity has scarcely been explored. Understanding FDs’ influence on health disparity especially concerns vulnerable groups, in terms of social class. Recent studies have expressed the opinion that primary health care might have a positive role in reducing health disparity, as the WHO has repeatedly argued. For example, O’Malley et al. noted that improving the quantity and quality of services offered in CHSCs was advantageous for reducing health disparities, reporting that people who received CHSC services were healthier, compared with those served by other medical institutions [34]. However, there are no direct investigations of the effect of FDs on health disparity or on the mechanisms by which this effect operates.

However, almost no research has been conducted on topics related to the effect of the FD system on health disparity for different SES groups. Specifically, several questions remain unanswered: (a) Has the FD policy eliminated health disparities in the past and how does the policy work over time? (b) Through what mechanisms has the FD policy contributed to the elimination of health disparity? (c) Would any other supportive policies be effective in eliminating health disparities? Each question is explored in this article.

## 2. Materials and Methods

### 2.1. Ethics and Approval

This study was approved by the Academic Ethics Committee of Shanghai Pudong Institute for Health Development (PDWSL2013-1). Participants gave written informed consent to participate in the research survey. Data were stored and processed anonymously.

### 2.2. Data and Measurements

The data used in this article were mainly from a tracking survey, and official data were obtained from the National Bureau of Statistics in China. Specifically, data on per capita health expenditure and per capita disposable income were acquired from the official statistics of the National Bureau of Statistics in China. Data on doctor visits, health behaviors, health outcomes, and registration with an FD were taken from a follow-up survey conducted by our research team. The tracking survey of permanent residents living in Shanghai, which mainly collected data on FD services, included information on basic demographic characteristics, medical treatment behavior, health behaviors, CHSC health service utilization, and health status. Multistage cluster sampling was used to identify a sample of 3040 residents (10 sub-districts * four Neighborhood Committees * two communities * 38 households * one resident per household). The first wave of the survey was conducted in 2013, when 40 trained investigators visited the selected residents, accompanied by Neighborhood Committee staff members. In 2016, 3 years later, a new set of investigators revisited the respondents. A total of 2754 valid questionnaires in 2013 and 2009 questionnaires in 2016 were collected.

The core variable of health was conceptualized in a comprehensive way. This study considered three dimensions of health status rather than specific diseases: subjective health (self-rated health), objective health status, and health behaviors. Self-rated health is a popular measure of health and is considered inclusive and accurate for determining health risk factors [35]. However, to consider objective physiological health and health behaviors as additional aspects of health status, indicators of non-communicable disease (NCD) status and several health behaviors were also included in our operationalization of health status. Thus, five indicators were used for health status in the present study: self-rated health, exercise frequency, smoking frequency, NCD status, and doctor-visiting behavior. The entropy method, an objective weighting method, was used to calculate weights for these five health indicators, and overall health status was obtained by multiplying the five health indicators by their weights. We also constructed a health gap variable by subtracting the mean comprehensive health indicator of the population with the highest SES from that of the population with the lowest SES.

SES was a core variable in this study. Previous research on the impact of SES on health disparity have mainly measured three aspects of SES: education, income, and occupation [36]. In China, the accuracy of income as a sensitive indicator that SES needs to be considered carefully. Several problems are presented by occupational division and classification, and income and occupation do not accurately represent living standards in the long term. Therefore, in the present study, we selected education as the main indicator of SES. Education was measured by asking “What is your education level?” (junior high school or less = 1, high school or secondary school = 2, undergraduate or higher = 3).

Other indicators were as follows: registration with an FD: “Have you signed up with a family doctor?” (yes = 1, no = 0); CHSC visiting behavior: “Do you visit a CHSC to see a doctor if you are sick?” (yes = 1, no = 0); tertiary hospital visiting behavior: “Do you go to a tier-2 or tier-3 hospital if you are sick?” (yes = 1, no = 0); rehabilitation status after visiting a CHSC: “Have you recovered after visiting a CHSC?” (yes = 1, no = 0); and rehabilitation status after visiting a tertiary hospital: “Have you recovered after visiting a tier-2 or tier-3 hospital?” (yes = 1, no = 0).

### 2.3. Model Construction

To explore the effects of the FD system on health disparity for different SES groups, this study established a dynamic system of health disparity using Vensim Simulation Software (Version 7.3.5). (Ventana Systems, Harvard, USA)

First, we mapped the causal loops of SES–health disparity using related theories of health disparity causes. Previous work has reported strong associations between life behaviors and SES, agreeing that higher SES groups tend to engage in healthier life behaviors, which result in better health status for these groups [37]. Further evidence has indicated that life behaviors are the mediating mechanism between SES and health status [38]. However, evidence has also demonstrated that people with higher incomes or levels of education find it easier to obtain high-quality medical resources [39]. Thus, two pathways were identified in our base model. The first was “SES–health behavior–health status” and the second was “SES–health resource seeking behavior–health status”. Three specific loops were developed, according to the base model. We called these loops the health behavior vicious loop, the doctor-visiting–health vicious loop, and the doctor-visiting restrain loop (see Figure 1).

The SES–FD–health disparity loop was then constructed. Based on the gatekeeper role of FDs, we included two intervention mechanisms of the FD system. The first was the health management mechanism, which is understood as the basic role of primary health care in population health [40]. Specific pathways were then developed, tracing the influence of the FD intervention on the health behaviors of people with different SES through health management to improve their lifestyles, breaking down the initial health behavior vicious loop. The second intervention mechanism was the cost control mechanism, which had an inhibitory effect on medical expenses [41]. With this intervention, medical expenses were reduced, the accessibility of medical resources was improved, and the rate of visits was increased, thereby forming a beneficial health condition improvement loop (see Figure 2).

In addition to the FD policy, we also introduced other policy interventions. We focused on two key points in the loops that might have a critical and comprehensive impact on the whole system. One point was reducing medical prices and the other was raising wages among the low-SES group. Previous studies have indicated that medical expenses are a significant factor affecting patients’ treatment and illness status [42]. The lack of access to medical resources caused by economic factors may also lead to the deterioration of health, which would have a negative impact on work and income, creating a vicious cycle. Thus, we introduced an intervention policy dealing with economic factors. We simulated the changes in health disparity after such policies were introduced, revealing the policy leverage point (see Figure 3).

Based on the causal loop, a more complicated dynamic model was constructed (see Appendix A
Figure A1), in which a game between primary health care institutions and tier-2/tier-3 hospitals was revealed, and such backbone was widely discussed in the healthcare system of China [43]. In this model, all variables and the relationships among them were quantified using equations, and causal loops of SES–health disparity, SES–FD–health disparity, and SES–FD–health disparity with policy interventions were deducted in the model respectively, which were named the business-as-usual (BAU) scenario, FD scenario, Policy 1 scenario (lower drug prices), Policy 2 scenario (a higher income level for the low-SES group), and policy hybrid scenario (with intervention policy 1 and 2). We divided the population into three SES categories by education (high, middle, and low levels), thus the health disparities were simulated and observed clearly over several years. We simulated this model with 2013 survey data, then we compared the fitted data of key outcome variables (including FD contraction rate, CHSC visit rate, tier-2/tier-3 visit rate, annual health expenditure per capita in CHSC or in tier-2/tier-3 hospitals, self-rated health, NCD, taking exercise, smoking, visiting a doctor if sick, and annual income per capita) with 2016 survey data and statistical data to test the stability of the model. We then further predicted system dynamics model results to the year of 2050, as we believed that the FD system was initially established but needed to be developed, but the impact of FD might be far-reaching. It is worth noting that, as education was used in dividing SES group categories, thus an education variable was not included in the figure, but an endogenous relationship of FD signing rate and different SES groups of people was considered, and we compared the predicted signing rate value with the 2016 survey as well. 

## 3. Results

### 3.1. Analysis of Health Status under the Family Doctor Policy Intervention

The business-as-usual (BAU) scenario showed that the health status of the more disadvantaged low- and middle-SES groups was significantly worse than that of the high-SES group, and the gap in health status between the best-off and worst-off was increasing (see Figure 4a). With the implementation of the FD policy, the health status of all SES groups improved to varying degrees. In the FD scenario, the health status of the low-SES group had the fastest growth rate, rising from 0.38 in 2013 to 0.66 in 2050. In 2039, the difference in health status between the low- and high-SES groups was projected to reach its smallest value—a difference of 0.040 (see Figure 4b). However, health disparities appeared to rebound after the health gap had been minimized in 2039. The health gap was projected to increase continually, then reach its peak level in 2050 (see Figure 4c).

We tried to find the main dominant factor of health disparity status and its trend. First, we found that changes in the hospital-visiting structure might have a positive effect on health status improvement. The system dynamics model showed that the hospital-visiting rate continued to decline for tier-2 and tier-3 hospitals, whereas the visiting rate for tier-1 care (i.e., CHSCs) continued to rise (Figure 5c,d), reflecting a shift in the former pattern of doctor-visiting behavior (Figure 5a,b). Specifically, the visiting rates in tier-2 and tier-3 hospitals for high-, middle-, and low-SES groups in the BAU scenario were approximately 82%, 70%, and 60% (Figure 5a), respectively, suggesting that higher-SES groups tended to visit large hospitals rather than CHSCs. In the FD scenario, the tier-2 and tier-3 hospital-visiting rates declined to 38%, 30%, and 25% in 2050 for the high-, middle-, and low-SES groups, respectively (Figure 5c).

A second explanation for the change in health status was that the effect of the percentage of income spent on medical costs largely cancelled out the positive effect of the medical treatment structure. Before the FD policy intervention, medical costs as a proportion of income increased over time for all SES groups (Figure 6a). Under the BAU scenario, medical costs were a much higher proportion of income for the low-SES group than for the other groups. The system dynamics model predicted that the percentages of income accounted for by medical costs would reach 16%, 35%, and 53% in 2050 for high-, middle-, and low-SES groups, respectively. After the FD policy intervention, the medical burden was somewhat reduced (Figure 6b), with the percentage of income spent on medical costs reaching its lowest value from 2030 to 2040. For example, the percentage of income accounted for by medical costs declined to 15.15% in 2032 for the low-SES group. However, the percentage then continually increased for this SES group, whereas it remained at low levels for the other SES groups (Figure 6b). FDs provide medical services at a relatively low price, which is a result of the cost control mechanism of the FD policy. However, for the low-SES group, who had lower income compared with the other groups, it was difficult to avoid a heavy medical burden, especially when the income growth rate was lower than the medical price growth rate. This led to what we call the treatment inhibition reinforcing loop.

### 3.2. Supporting Policy Intervention Simulation

After the FD policy intervention, the proportion of income spent on medical costs remained high, resulting in the rebound of the health gap, which was our primary area of concern. Thus, we introduced two supporting policy interventions: lower drug prices and a higher income level for the low-SES group.

We found that the measure to decrease medical costs would have little effect on reducing health disparity. The policy of lowering drug prices reduced medical costs for all SES groups, but it did not have a significantly stronger effect for the lower-SES groups. Thus, the disparity in the percentage of income spent on medical costs among different SES groups remained apparent. This percentage was projected to reach its lowest value for the low-SES group in 2036 (12.31%); afterwards, the percentage was projected to continually increase, while it decreased for the other groups (Figure 7a). However, the rate of visiting a doctor when sick was projected to increase among all SES groups, with a great reduction in the gap between the different groups. From 2023 to 2045, the doctor-visiting rate of people with low SES was even predicted to exceed that of the middle-SES group. After 2045, the doctor-visiting rate of the low-SES group showed a downward trend and the disparity between those with low SES and the other SES groups expanded again (Figure 7b). The health gap between the high- and low-SES groups was projected to decrease from 0.17 in 2013 to 0.0363 in 2040, but then to widen again after 2040 (Figure 7c). This expansion of the health gap indicated the absence of a sustained effect to the policy in terms of reducing health disparity.

Compared with the policy to reduce medical prices, raising the income level for the low-SES group greatly relieved the medical burden. Here, the proportion of income spent on medical costs was projected to decline sharply after an initial drop from 2021 to 2027, reaching 6.53% in 2050 (Figure 8a). At the same time, this policy was projected to improve the downward trend in the doctor-visiting rate. By 2050, the rate of visiting a doctor when sick would reach 0.670 for the low-SES group, which is higher than the projected rates for the middle- and high-SES groups (Figure 8b). The system dynamics simulation results showed that this supporting policy effectively suppressed the rebound of the health gap, which stabilized at around 0.03 (Figure 8c).

### 3.3. Policy Conparison

To further understand the comprehensive intervention effect of the FD policy and supporting policies on health disparity, three policies (the FD policy, lower medical prices, and a higher income level for the low-SES group) were introduced simultaneously in the model. On the whole, the results indicated that the simultaneous implementation of these three policies could greatly improve the level of health for the different SES groups, although health status still remained worse for the low-SES group than for the other two groups. Specifically, health status was projected to rise from 0.378 to 0.685 in the low-SES group, from 0.501 to 0.696 in the middle-SES group, and from 0.554 to 0.715 in the high-SES group (Figure 9a). Projected medical costs for all three SES groups showed a downward trend after 2025, and this trend was especially strong for the low-SES group. By 2050, the proportion of income spent on medical costs by the low-SES group would fall to 0.037 (Figure 9b). An apparent improvement in the doctor-visiting rate was also observed for all three SES groups, especially for the low-SES group, which was projected to reach a level comparable to that of the high-SES group (Figure 9c).

We also compared the health gap under different scenarios. The system dynamics model showed that the FD policy, the medical price reduction policy, and the low-income improvement policy all had certain mitigation effects on health disparity, and the minimum values of the health gap between the low- and high-SES groups were 0.0404, 0.0363, and 0.0267 for the three policies, respectively. Under the combined intervention of the three policies, the minimum health gap was 0.0259 (Figure 10). The health gap increased to varying degrees after reaching its minimum value, as can be seen clearly in Figure 10. The simulation model suggested that the low-income improvement policy was more significant in terms of reducing the health gap based on the FD scenario.

## 4. Discussion

Our results showed obvious health disparities between people with different SES. People with higher SES had better health status, which was highly consistent with the findings of previous work. For example, Kennedy et al., who also defined three SES groups (low, middle, and high), found that self-reported diabetes, hypertension, heart disease, obesity, and strokes were highest in the lowest-SES group [44]. Previous studies conducted in China have also revealed health disparities among different SES groups [45].

Another important finding of the present study was that the FD policy intervention improved the health status of all three SES groups and narrowed the health gap between them, especially the gap between the high-SES group and the low-SES group. This is one of the first studies to use empirical methods to show the positive effect of the FD policy on health disparity. Many earlier studies have discussed the effect of primary health care on improving population health, reaching consensus that primary health care has a positive effect on health improvement. For example, a study conducted by Macinko and colleagues revealed an association between the primary care physician supply and improved health outcomes, including all-cause, cancer, heart disease, strokes, and infant mortality; low birth weight; life expectancy; and self-rated health [46]. However, no previous research has pursued the precise quantitative analysis of the effect of the FD system on health disparity, although the WHO has widely advocated the potential of primary health care for promoting health equity. Previous work has provided indirect support for the link between primary health care and the reduction of inequalities in health. Starfield et al., for example, argued that cumulative contributions by primary health care could reduce the problem of health disparity [30], and Shi et al. put forward similar opinions, asserting that the FD system was associated with lower mortality and might even eliminate the adverse influence of health on income inequality [47].

The system dynamics simulation model results indicated that the medical cost control mechanism had a stronger effect on the system. The FD policy affected the doctor-visiting behavior of people with low SES. The system dynamics model predicted that, with the FD policy intervention, people would seek health resources more frequently when sick and be more likely to first visit CHSCs rather than tier-2 or tier-3 hospitals, compared with the situation before this policy intervention. Furthermore, these changes were predicted to be most obvious for people with low SES. Additionally, medical costs were predicted to be controlled to some degree after the change in doctor-visiting behavior. One significant indicator was the percentage of income spent on medical costs, which declined significantly after the implementation of the FD policy, especially for people in the low-SES group.

The mechanism uncovered in this study is quite reasonable, given that, although the FD system is new, it was originally proposed more than 50 years ago. “Barefoot doctors”, a significant inspiration for primary health care [26], aided in the distribution of health care resources under an urban–rural dual economic structure, and the WHO has referred to this earlier system as a successful example of managing shortages in medical resources in a developing country [25]. Primary health care was not re-recognized or valued again until 2009, with the new health care reforms. The FD system has been established over the last 10 years. During this period, the government has tried to encourage residents to register with and visit FDs by offering lower medical costs. However, a common criticism is that residents were automatically registered with FDs, potentially leading to an overestimation of the registration rate [48]. The government, recognizing this problem, changed its policy target from the achievement of a certain registration rate to a goal tied to FD service provision. Clearly, the health management function of the FD system remains far from being realized, and we believe that the health management mechanism will play an increasingly significant role in improving health disparity with the further development of the FD system in China.

Our results indicated that the FD policy will not be able to eliminate health disparity in the long run, although it was projected to have positive effects in its early stages. The system dynamics simulation results predicted that the health gap among different SES groups would gradually be eliminated over time under the FD policy intervention, but a rebound in the health gap was then predicted to appear after this disparity reached its lowest value. After introducing two supportive policies, we found that the medical price control policy for the entire population would fail to eliminate health disparity, but the income improvement policy, which was based on SES itself, would be more effective, suggesting that SES might be the fundamental factor influencing health disparity. In fact, similar findings in previous work have resulted in a broad consensus among scholars regarding the fundamental nature of SES in determining health disparity [49]. It can be inferred that health disparity cannot be completely eliminated, as long as disparities in SES persist. The relationship between social class and health disparity merits further exploration in future studies.

## 5. Limitations

There are several limitations of this study. Firstly, in order to test the stability, we only used the 2016 survey data to compare with the predicted results of the 2013 survey data fitted in the model, as the largest sample survey was only conducted in 2013 and 2016. It could be more rigorous if more waves of data are compared with the predicted data. Secondly, factors of population age structure, general socio-economic system, and cultural and living conditions were neglected in this study, which are significant predictors of health. Thirdly, this study was conducted in Shanghai, the most advanced metropolitan in China, the results of which might not apply in other regions, especially the remote rural areas in China, indicating a comparison research in primary health care and health disparity of urban and rural areas worth studying. Lastly, although our theory model (causal loops) was constructed according to classic researches, it is better to construct the system dynamics model, especially the model structure based on current studies, which will be more convincing.

## 6. Conclusions

The FD policy was found to play a positive role in reducing health disparity in Shanghai, China. Compared with health management, the reduction of medical costs was found to be a more important mechanism, through which the FD system affected health among different SES groups because the realization of the health management function of the FD system remains distant. We further agree with the widely accepted conclusion that SES is the fundamental cause of health disparity. Consequently, the FD policy cannot completely eliminate health disparity.

## Figures and Tables

**Figure 1 ijerph-17-05548-f001:**
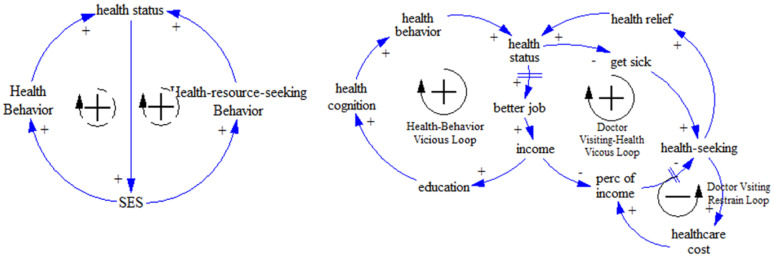
Causal loop of socioeconomic status (SES)–health disparity.

**Figure 2 ijerph-17-05548-f002:**
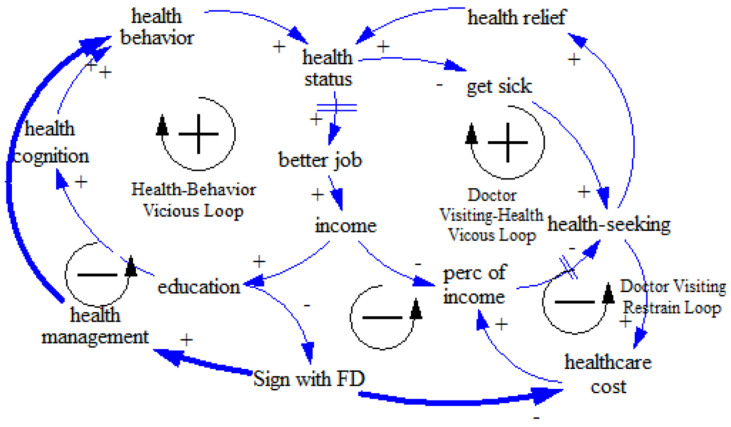
Causal loop of SES–family doctor (FD)–health disparity.

**Figure 3 ijerph-17-05548-f003:**
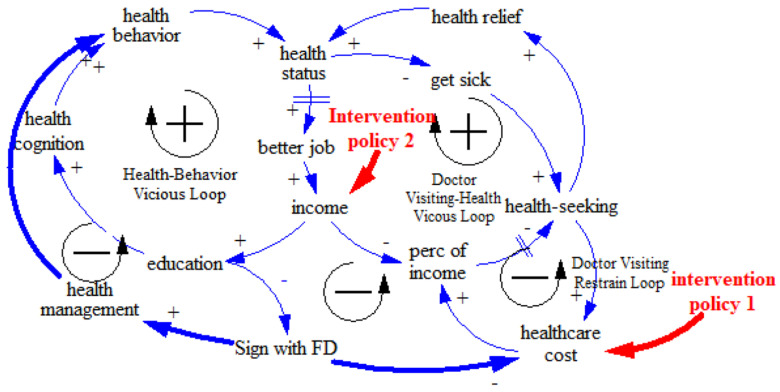
Causal loop of SES–FD–health disparity with policy interventions.

**Figure 4 ijerph-17-05548-f004:**
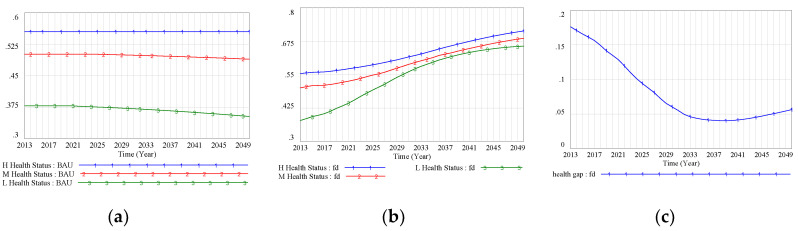
(**a**) Health status (business-as-usual (BAU) scenario); (**b**) health status (FD scenario); (**c**) health gap (FD scenario). Note: BAU scenario: SES–health disparity scenario; FD scenario: SES–FD–health disparity scenario; Policy 1 scenario: scenario with intervention policy of lower drug prices; Policy 2 scenario: scenario with intervention policy of higher income promotion for the low-SES group; Policy hybrid scenario: scenario with two intervention policies.

**Figure 5 ijerph-17-05548-f005:**
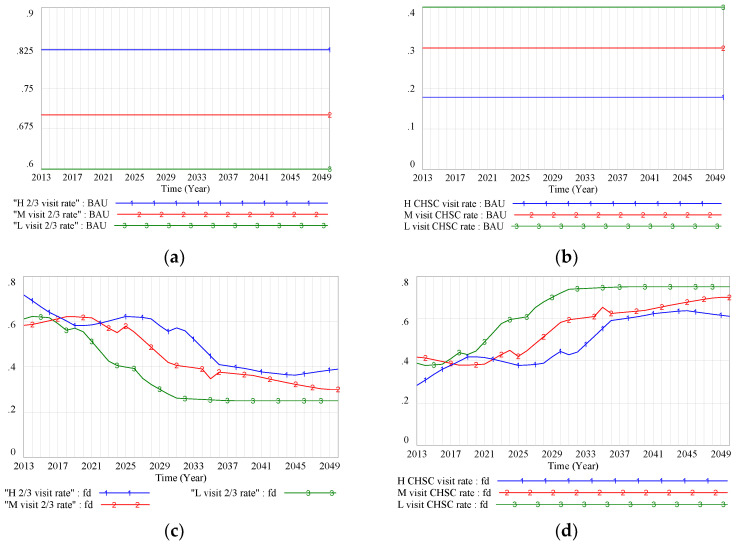
(**a**) Tier-3 hospital-visiting rate (BAU scenario); (**b**) community health service center (CHSC)-visiting rate (BAU scenario); (**c**) tier-2/tier-3 hospital-visiting rate (FD scenario); (**d**) CHSC-visiting rate (FD scenario). Note: BAU scenario: SES–health disparity scenario; FD scenario: SES–FD–health disparity scenario; Policy 1 scenario: scenario with intervention policy of lower drug prices; Policy 2 scenario: scenario with intervention policy of higher income promotion for the low-SES group; Policy hybrid scenario: scenario with two intervention policies.

**Figure 6 ijerph-17-05548-f006:**
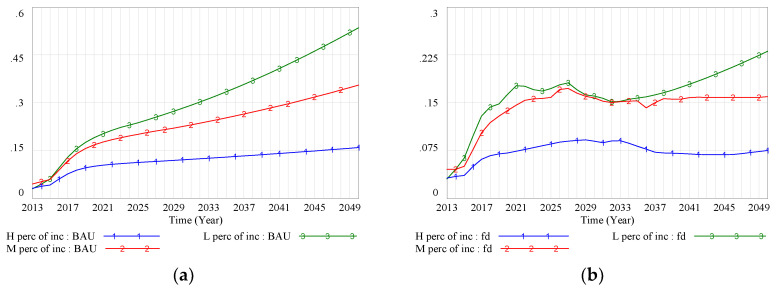
(**a**) Medical costs as a proportion of income (BAU scenario); (**b**) medical costs as a proportion of income (FD scenario). Note: BAU scenario: SES–health disparity scenario; FD scenario: SES–FD–health disparity scenario; Policy 1 scenario: scenario with intervention policy of lower drug prices; Policy 2 scenario: scenario with intervention policy of higher income promotion for the low-SES group; Policy hybrid scenario: scenario with two intervention policies.

**Figure 7 ijerph-17-05548-f007:**
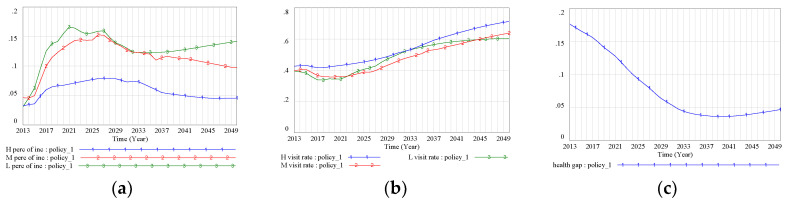
(**a**) Medical costs as a proportion of income (Policy Scenario 1); (**b**) doctor-visiting rate (Policy Scenario 1); (**c**) health gap (Policy Scenario 1). Note: BAU scenario: SES–health disparity scenario; FD scenario: SES–FD–health disparity scenario; Policy 1 scenario: scenario with intervention policy of lower drug prices; Policy 2 scenario: scenario with intervention policy of higher income promotion for the low-SES group; Policy hybrid scenario: scenario with two intervention policies.

**Figure 8 ijerph-17-05548-f008:**
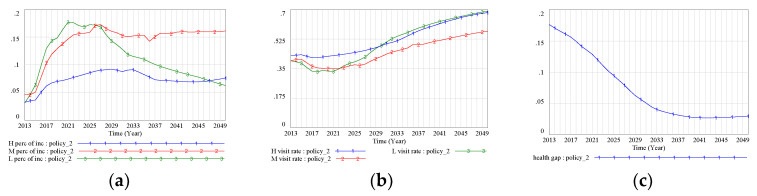
(**a**) Medical costs as a proportion of income (Policy Scenario 2); (**b**) doctor-visiting rate (Policy Scenario 2); (**c**) health gap (Policy Scenario 2). Note: BAU scenario: SES–health disparity scenario; FD scenario: SES–FD–health disparity scenario; Policy 1 scenario: scenario with intervention policy of lower drug prices; Policy 2 scenario: scenario with intervention policy of higher income promotion for the low-SES group; Policy hybrid scenario: scenario with two intervention policies.

**Figure 9 ijerph-17-05548-f009:**
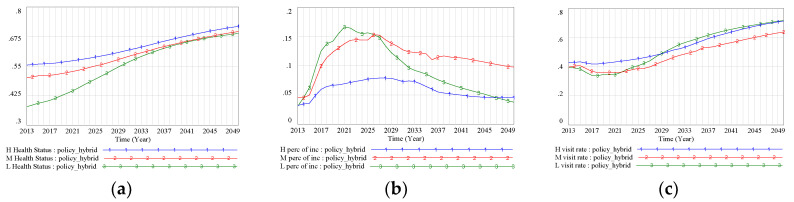
(**a**) Health status (Hybrid Scenario); (**b**) medical costs as a proportion of income (Hybrid Scenario); (**c**) doctor-visiting rate (Hybrid Scenario). Note: BAU scenario: SES–health disparity scenario; FD scenario: SES–FD–health disparity scenario; Policy 1 scenario: scenario with intervention policy of lower drug prices; Policy 2 scenario: scenario with intervention policy of higher income promotion for the low-SES group; Policy hybrid scenario: scenario with two intervention policies.

**Figure 10 ijerph-17-05548-f010:**
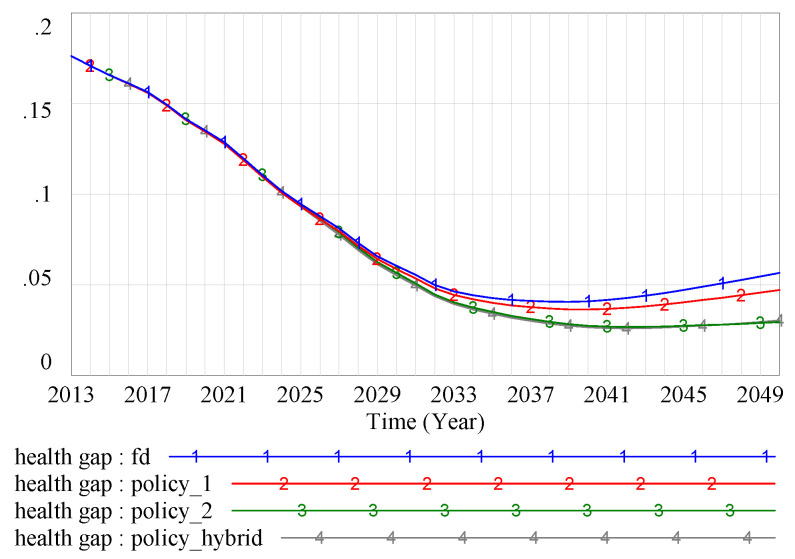
Health gap under different scenarios. Note: BAU scenario: SES–health disparity scenario; FD scenario: SES–FD–health disparity scenario; Policy 1 scenario: scenario with intervention policy of lower drug prices; Policy 2 scenario: scenario with intervention policy of higher income promotion for the low-SES group; Policy hybrid scenario: scenario with two intervention policies.

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
