# Peer review of "Can Health Disparity Be Eliminated? The Role of Family Doctor Played in Shanghai, China"

_ijerph, 2020, doi:10.3390/ijerph17155548_

Round 1

Reviewer 1 Report

This is an important piece of work that addresses a very important problem (health disparities as driven by SES) in the most dynamic economy in the work (China).  The major empirical method appears to be survey methods, but this particular paper seeks to leverage those empirical results by extrapolating the work using a system dynamics model.  It is in this extrapolation using system dynamics models that I have most of my comments.

The key issue with this paper centers on Figure 4.  As a reader, while I am trained in system dynamics modeling, I cannot grasp fully how the model presented in this figure was constructed.  The text sets up interesting survey methods completed in Shanghai and then posits several feedback effects in Figure 3.  Then comes Figure 4 more or less as a surprise.  It is difficult for me to exactly unravel how the feedback described in Figure 3 is formulated and then calibrated in Figure 4.  It looks to me as if there are significant exogenous effects within the model.  These might enter in two ways.  First, the HFD, MFD, and LFD variables appear to be direct time functions.  Second it appears that these variables may feed through some form of table functions involving smoking, exercise, and chronic disease to introduce additional exogenous dynamics.  So it is hard for me to sort out endogenous behaviors from exogenous behaviors in the presented conclusions.

In addition, the main stock and flow dynamics in each of the three "sectors" appears to be triggered by some kind of hospital event and the stock and flow structure then seems to "sort out" the population using a split flow structure with no "return flows" shown.  The model seems to show permanent residents who flow into the system and then leave the system via six "recovery" rates associated with 2/3 (secondary? and tertiary?) hospitalization and CHSC (tier 1 primary care treatments?).  This backbone structure needs to be more carefully explained, justified, and related to similar mechanisms in the existing literature.

The "fix" is to be much more careful in describing the formulation and calibration within your actual model so that the reader can judge for herself how you have set up the model.  This is hard to do within the format of this publication and you might want to get this documentation and review completed in one of the "systems" journals and then just cite that paper indicating to the reader where this review of formulation and calibration took place.

Another approach that you should consider would be to tie your structure to already reviewed solid work, demonstrating to the reader how your work is an extension of existing work that has already gone through the tough job of peer review of structure and behavior with its associated calibration to data processes.  While my field is not this kind of health care work, I think that the best work in the system dynamics field (the field that you seem to want to be working in) is being done by Jack Homer, Gary Hirsch, and Bobby Milstein (especially all of the work that they are doing in the "ReThink Health"area).  You should certainly cite that good work and relate this work to it. Also, there is some recent and very good work on primary care that relates to this paper that has been done by Andrada Tomoaia-Cotisel currently working out of RAND.

This is important work that you are doing.  Health disparities are very important and you bring much needed research and perspective from the Chinese experience.  This is important work that needs to be published.

Reviewer 2 Report

Greetings:

Please check the article for grammatical errors and incomplete sentences e.g. line 9.

You might want to consider explaining the role of family docs versus primary care docs and services. 

You have provided detailed information using systems thinking simulation. Who is your reader? Individuals without a background or knowledge in your simulation software may not appreciate the elaborate exercises or findings that you put forth. Luckily, you have provided some explanation of the findings which unfortunately were not different from other similar studies undertaken. 

Reviewer 3 Report

Thank you for the change to review the interesting manuscript about how Family Doctor System may be closing the health gap between better and worse off. This is a very important topic, and relevant for wide variety of audiences. The authors have also used innovate methods in addressing their research questions. Currently the major weakness in the manuscript is introduction section, which appears to miss key information or presents the information in incomplete form. This makes following the rest of manuscript harder. Detailed feedback is included under.

Abstract

Line 10: „Primary health care that..” – Please check the sentence. Word “that” should be removed.

Line 12 / methods: Consider rewriting, based on the information provided, it is not possible to form an idea what methods were used to evaluate the impact. What kind of framework was used? The dynamics system? Which techniques did this involve?

Line 15: “…would play…” please could the abstract define more carefully at the beginning whether the evaluation is concerning the last 10 years or offers also projections for the future.

Introduction

Line 26: Marmot studies/report is influential in highlighting socio-economic differences. However, the sentence is very difficult to understand unless the reader is already familiar with the report. For example, it may not be clear what civil servants are. Authors should consider expanding the text including shortly explaining how social gradient influenced health, i.e. what differences in health could be seen between social gradients.

Line 27: Likewise with the Black Report, the sentence is difficult to follow and authors should consider adding some details.

Line 30: “Subsequent…” Please consider shortly adding few details, what disparities the studies show.

Line 32: Health disparities have increased… Which timeframe this refers to? i.e. last 20 years? 30 years?

Line 35: “different classes” – considering the authors emphasis on social classes, the introduction is difficult to follow when the term “social class” and different social classes as referred by the authors are not clearly defined.

Line 39:”…countries of former Soviet Union…” Please specify which countries.

Line 43: “A range of resources…” Authors should consider introducing this information earlier in the induction.

Line 48 “Primary health care has been highlighted, in contrast to the traditional approach to health care “. Where or who has highlighted primary healthcare and why is primary healthcare been highlighted (what is traditional approach to health care and is this uniform around the world?

Line 52: “policy” – does this refer to national policies or WHO (usually national governments do policies and WHO recommendations)?

Line 58: ”…collective labor…” in what context? Collective medical labor in primary health care?

Line 61: “great” – please consider using “considerable”

Line 69: “Healthy China 2030” – Is this a policy or an initiative?

Line 70: Please, could you shortly explain what a family doctor system in this context is?

Line 92: “Primary health care in China originated with the barefoot doctor program, which has been reconstructed as the FD system since 2013.” Repeat information, please consider if needed.

Line 95: The questions a) & c) appears in fact as being two different questions. Also, authors should consider the language use in the research question. E.g. “universally eliminated…” Was that the stated aim of the policy or rather improved/lessened the disparities? In addition, the aims do not clearly mention the simulation work done as a part of the analysis.

Methods

Please could you confirm whether participants gave written informed consent to participate in the research survey or whether another mechanisms was used.

Line 114: “The first wave of the survey was conducted in 2013, when 40 trained investigators visited the selected residents, accompanied by Neighborhood Committee staff members. Three years later, in 2016, a new set of investigators revisited the respondents. A total of 2754 valid questionnaires in 2013 and 2009 in 2016 were collected.” Please could you clarify – were there 2 or 3 measurement points?

Line 148: “…using theories…” Please define which theories. Have these already been introduced in the induction?

Results

Figure 4 – comprehensive but very difficult to follow due the size and complexity.

For example Figures 8 – the time scale is not same in all figures, and should be either clarified or corrected. In general, for readers less familiar with this type of analyses, a short explanation should be added about how the results are interpreted. Especially as scale used to denote the results differs between the figures, making comparison between the figures harder.

It would also be good to discuss (in the method section) how was the timepoint for modelling decided (e.g. around 2050). What is the authors’ judgement how reliable the predictions based on their data are?

In addition, there is duplication in the results section, especially when results are presented both in numeric and graphical form. This makes also the results difficult to follow. Authors should consider whether it would be possible use e.g. tables to make results easier to follow.

Further, regarding the understanding the figures, it should be considered to mark more clearly which line denotes which SES. H/M/L may be difficult to understand.

Considering the complexity of the analyses, it should be considered to add in the captions e.g. what was the policy scenario two e.g. figures 9.

Discussion

Authors should consider the limitations of the study and which factors not included in the model (e.g. environmental factors, population structure etc.) may interact with SES and health.

Also, the research was conducted within a large metropolitan area. Authors may want to consider discussing whether results are applicable outside of this context.

Round 2

Reviewer 3 Report

Thank you having the change to review the manuscripts after revisions. The authors have considered all the suggested changes and in my opinion the manuscript has improved considerably.

There are few further points, mainly related to revisions that the authors have done were there clarity of the text should be improved.

Abstract

Line 13: Please define what is BAU

Introduction

Line 29: Thank you for adding the details. However, the term ”civil servants” may still not be clear for all readers. This is clarified later as “(administrators) - Please could you clarify this earlier in the text.

Line 46: “Most scholars consider SES to be a key factor affecting health.” Please could you add reference.

Line 55-63: New added text. Please could break this in smaller sentences, currently this is difficult to follow.

Line 87-93: Please could you break this to smaller sentences and check the grammar. Currently this is difficult to follow.

Line 205: “…(see appendix)…” Please could you revise this sentence as in its current form its meaning is very difficult to understand.

Line 213: “SES categories by education (high, middle, and low levels)…” Were these categories defined previously? If not please could you add.